



# Seismic waveform tomography of the Central and Eastern Mediterranean upper mantle

Nienke Blom[1,3], Alexey Gokhberg[2], and Andreas Fichtner[2]

[1]Bullard Laboratories, Department of Earth Sciences, University of Cambridge, Madingley Rise, Cambridge CB3 0EZ, United Kingdom
[2]Department of Earth Sciences, ETH Zürich, Sonneggstrasse 5, 8092 Zürich, Switzerland
[3]previously: Department of Earth Sciences, Universiteit Utrecht, Princetonlaan 8A, 3584 CB Utrecht, Netherlands

**Correspondence:** Nienke Blom (nienke.blom@esc.cam.ac.uk)

**Abstract.** We present a seismic waveform tomography of the upper mantle beneath the Central and Eastern Mediterranean down to the mantle transition zone. Our methodology incorporates in a consistent manner the information from body and multimode surface waves, source effects, frequency dependence, wavefront healing, anisotropy and attenuation. This allows us to jointly image multiple parameters of the crust and upper mantle.

Based on the data from ∼17 000 unique source-receiver pairs, gathered from 80 earthquakes, we image radially anisotropic S velocity, P velocity and density. We use a multi-scale approach in which the longest periods (100–150 s) are inverted first, broadening to a period band of 28–150 s. Thanks to a strategy that combines long-period signals and a separation of body and surface wave signals, we are able to image down to the transition zone in most of the model domain.

Our model shows considerable detail in especially the northern part of the domain, where data coverage is very dense, and
displays a number of clear and coherent high-velocity structures across the domain that can be linked to episodes of current and past subduction. These include the Hellenic subduction zone, the Cyprus subduction zone and high-velocity anomalies beneath the Italian peninsula and the Dinarides. This model is able to explain data from new events that were not included in the inversion.

## 1  Introduction

Since the late 1970s (Dziewoński et al., 1977; Aki et al., 1977), seismic tomography has emerged as the primary method for imaging the Earth's interior from the kilometre to the global scale. On regional to continental scales, the 3-D images can help to decipher the tectonic situation and history of an area by linking the surface observations to structures deeper in the mantle: high-velocity structures have been associated with subduction since the earliest days of seismic tomography.

The Mediterranean in particular is an area that has attracted much attention from the beginning – not in the least because
of its strong seismicity and good data coverage. Classical ray tomography has been applied numerous times to study this area, both using body and surface waves (e.g. Spakman et al., 1988; Piromallo and Morelli, 1997, 2003; Amaru, 2007; Biryol et al., 2011; Koulakov et al., 2015; Portner et al., 2018; Snieder, 1988; Zielhuis and Nolet, 1994; Marone et al., 2004; Schivardi and Morelli, 2009; Salaün et al., 2012; Legendre et al., 2012).





As data availability increased, structures have thus been imaged with increasing amounts of detail. In the past, this has already
led to hypotheses on the kinematics of the Mediterranean region that were testable against independent surface geological data
(see e.g. Wortel and Spakman, 2000; Faccenna et al., 2014). Nevertheless, ambiguities remain: in particular the differences
between body and surface wave studies can be striking, where surface wave anomalies are generally of larger amplitude, but
laterally smoother. Another difficulty is the connection between the crust and mantle, with most methods relying on some input
crustal model whose accuracy may vary, depending on location (e.g. Bozdağ and Trampert, 2008; Laske et al., 2013; Fichtner
et al., 2013b).

Continuing improvements in computing power have meanwhile facilitated the development of waveform tomography tech-
niques (also referred to as full-waveform inversion or adjoint tomography), that allow us to fully account for the physics of
wave propagation in 3-D heterogeneous media. By directly comparing observed seismograms with a simulated 3-D wave-
field, a wealth of information can be extracted from the data. While computationally more expensive than ray-based imaging
methods, the advantage of waveform methods lies in their ability to incorporate in a consistent manner all the information in
seismograms – not just the arrivals of certain, specified phases. As a result, body and multimode surface waves, source effects,
frequency-dependence, wavefront healing, anisotropy and attenuation are naturally and coherently incorporated. The use of
such a forward and inverse modelling technique largely excludes modelling artefacts in the imaging. Issues like the contamina-
tion of mantle structure that results from insufficient crustal corrections can thus be avoided (e.g. Montagner and Jobert, 1988;
Bozdağ and Trampert, 2008). This makes waveform tomography especially suitable for imaging tectonically active parts of the
Earth such as the Mediterranean, where large contrasts in elastic properties are likely to be present. It thus allows us to image
S-wave and P-wave velocity jointly for the crust and mantle, but also makes it possible to put constraints on density that ray
tomography cannot provide.

## 1.1 Objective and outline

In this study, we use waveform tomography to image the upper mantle beneath the Central and Eastern Mediterranean, inverting
for the seismic velocities $v_{\mathrm{SH}}$, $v_{\mathrm{SV}}$ and $v_{\mathrm{P}}$ as well as density $\rho$. Compared to the earlier, larger scale European model of Fichtner
et al. (2013b), our study provides a more local and zoomed-in update. In the present study, we pay particular attention to
developing a strategy that optimises the sensitivity to the deeper parts of the model domain by using long-period data and, at
shorter periods, a windowing strategy optimised for extracting body wave signals. This manuscript focuses on the technical
construction of the model, discussing in detail the used methodology and the uncertainties and caveats of the methods and
results. While we will highlight some key features of the resulting model, a detailed geological discussion will be the subject
of a follow-up study.

After a brief introduction to the geological setting of the Mediterranean (Section 2), we will focus on the methodological
aspects of the waveform tomography executed, discussing the model domain and data selection (Section 3) as well as the
inversion setup (Section 4). The latter includes a description of the effects of window selection on the inversion. In Section
5, we discuss the main features of the resulting model, after which Section 6 is dedicated to validity tests of the model and a
discussion of the limitations of the methodology.





## 2 Geological setting

Slow convergence between the African and Eurasian plates dominates the geological and tectonic setting of the Mediterranean
domain (McKenzie, 1972; Dewey et al., 1989; Wortel and Spakman, 2000; Faccenna et al., 2014). This convergence (approx. 6
mm/yr, Reilinger et al., 2006) is currently mainly accommodated in the Alpine arc, the Hellenic Arc and through complicated
interactions between several plates and microplates present in the domain.

The Mediterranean is comprised of two basins of contrasting characteristics. The Western Mediterranean consists of young
oceanic lithosphere (∼30 Ma and younger, Wortel and Spakman, 2000), and the Central/Eastern Mediterranean is predomi-
nantly old (up to 340 Ma, Granot, 2016). This dichotomy is ultimately a result of the collision of the Adria plate (a promontory
of the African plate) with the Eurasian crust at the location of the Alps – effectively cutting the basin in two (Channell, 1996;
Faccenna et al., 2014). The young Western Mediterranean formed during subsequent opening of the Liguro-Provençal and
Tyrrhenian basins as a result of trench roll-back (Faccenna et al., 2004). This rotated what is now the Italian peninsula to its
current NW–SE orientation and resulted in the formation of the Apennines. It also brought Corsica and Sardinia from Iberia
to their current locations and resulted in the steeply dipping Tyrrhenian subduction zone (e.g. Spakman and Wortel, 2004;
Koulakov et al., 2015). The Central/Eastern Mediterranean, on the other hand, consists mainly of old African oceanic litho-
sphere – with the exception of the Aegean Sea, where current subduction beneath the Hellenic Arc and trench roll-back towards
the south-west and south have resulted in a young extensional basin (e.g. McKenzie, 1978; Jolivet et al., 1994).

Several relatively large microplates play significant roles in the system (Figure 1a). On the eastern end of the Mediterranean,
the Arabian plate is moving northwards towards Eurasia along the Dead Sea Fault. West of this, and accommodated by the
North Anatolian Fault and East Anatolian Fault, the Anatolian microplate moves westwards relative to Eurasia and Arabia.
Continuing west, the Aegean microplate moves in a SW direction relative to Eurasia. The combination of these motions results
in a counterclockwise rotation that has been attributed to the rollback of the Hellenic slab and effects of mantle convection
and gravitational potential energy as a result of Anatolia's high elevation (Faccenna et al., 2014). Another subduction zone is
visible beneath Cyprus, which some consider to be separated from the Hellenic subduction through the formation of a STEP
fault (Govers and Wortel, 2005; Özbakır et al., 2013).

## 3 Model domain and data

### 3.1 Choice of model domain

The chosen study area covers most of the Central and Eastern Mediterranean. This includes the tectonically interesting regions
of the Italian peninsula, the Hellenic Arc and Anatolia, but also stretches towards the south to include the African coast (Figure
1). In order to avoid artificial reflections from the boundaries when simulating wave propagation within this model domain, a
buffer zone with absorbing boundaries is implemented (Cerjan et al., 1985).





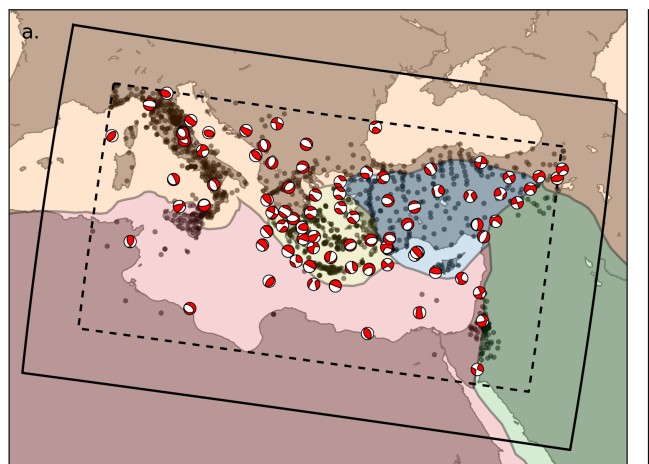
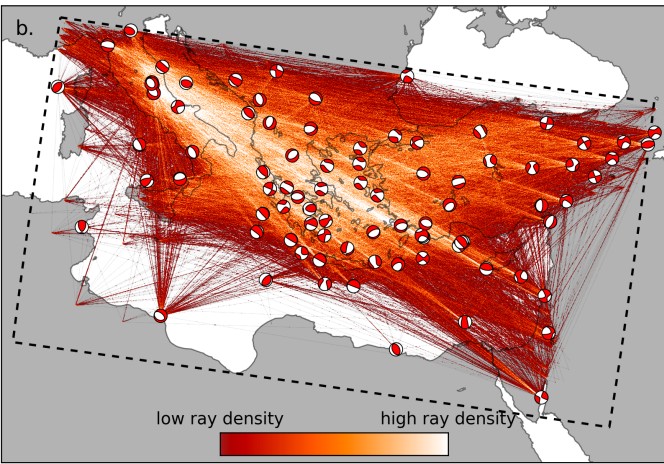

**Figure 1.** (a) The modelling domain in the Central and Eastern Mediterranean, with tectonic plates taken from Bird (2003): Africa in red, Eurasia in orange, Aegean in yellow, Anatolia in blue and Arabia in green. Superimposed on top of this are the earthquakes used in this study (red-white focal mechanisms), and the locations of all seismic stations (black dots). A 3° buffer zone separates the outer and inner model boundaries (solid and dashed lines, respectively). Within the buffer zone, wave propagation energy is absorbed that would otherwise result in artificial reflections. (b) An impression of "ray density" in the model domain, based on the great circle paths of all traces used in this study. This is just a rough proxy of coverage, serving only to highlight the variability and directionality of the coverage.

## 3.2 Data selection

The inversion is carried out using data from around 80 earthquakes that occurred within the model domain between 1998
and 2017 (Figure 1a, Supplementary Table 1). Most of the tectonic activity is in the north of the model domain, so in order to obtain a coverage that is as homogeneous as possible, events are initially selected manually from the IRIS SPUD moment tensor catalog (http://ds.iris.edu/spud/momenttensor). Additional events are then obtained using automatic event selection from the Large-scale Seismic Inversion Framework package (LASIF; Krischer et al., 2015, http://lasif.net).

All moment tensors are taken from the Global CMT Project (Ekström et al., 2012; Dziewonski et al., 1981, http://www.
globalcmt.org). Seismograms are downloaded automatically from the IRIS, ORFEUS, ETH, GFZ, and KOERI data centres using the LASIF package. Since especially the depth and the moment tensor components linking horizontal and vertical motions can be poorly constrained when determined from long-period data (e.g. Jackson, 2001), we manually monitor the suitability of the earthquake data at all stages during the inversion. We excluded events that provided too few reliable measurements, e.g. because of cycle skipping or noise issues. In a few cases, we relocated or changed the timing of events where the (spatial)
distribution of phase shifts indicated a clearly interpretable pattern (see Section 2 of the Supplementary material).

This approach results in around 80 events at the beginning of the inversion, corresponding to about 17 000 event-station pairs or 50 000 single-component channels, with a coverage that is excellent in the north but limited in the south (Figure 1b). A table containing all earthquakes is available in Supplementary Table 1.





## 4 Inversion setup

We perform regional waveform tomography by comparing the downloaded seismograms to synthetic seismograms, computed for subsequent iterations of model updates. Our inversion uses a deterministic, gradient-based iterative approach, such that the synthetic seismograms for consecutive models progressively provide a better match with the observed data. This optimisation method is local: updates are obtained by continuously moving in a direction of descent. There is therefore no guarantee that it will descend towards the global minimiser, i.e. the (set of) model(s) that would result in the lowest total misfit. The resulting

model is therefore a function of several important strategies and choices, which are discussed below.

### 4.1 Seismic wave propagation

Synthetic seismograms are computed by simulating seismic wave propagation in 3-D within the model domain. Here, we solve the wave equation

$$\rho(\mathbf{x})\,\ddot{\mathbf{u}}(\mathbf{x},t) - \nabla \cdot \boldsymbol{\sigma}(\mathbf{x},t) = \mathbf{f}(\mathbf{x},t), \tag{1}$$

where $\rho$ denotes density, $\mathbf{u}$ is the displacement field ($\ddot{\mathbf{u}}$ being its second time derivative the acceleration field), $\boldsymbol{\sigma}$ the stress field and $\mathbf{f}$ the forcing term. We use the SES3D wave propagation code (Fichtner and Igel, 2008; Gokhberg and Fichtner, 2016) and invert for a rheology that includes radial anisotropy for S-wave velocity (SH and SV). Anelasticity, kept fixed, is implemented using memory variables (Blanch et al., 1995; Fichtner and van Driel, 2014). The rheology relates stress $\boldsymbol{\sigma}$ to displacement $\mathbf{u}$.

### 4.2 Misfit definition

In order to quantify the differences between observed and synthetic waveforms, we use the time-frequency phase misfit functional of Fichtner et al. (2008)

$$J_{\mathrm{TF},p}(\mathbf{m}) := \int_t \int_\omega W_p^2(t,\omega)\left[\phi(\mathbf{m},t,\omega) - \phi_{\mathrm{obs.}}(t,\omega)\right]^2 \mathrm{d}t\mathrm{d}\omega,$$

$$W_p = \frac{\log(1+|\tilde{u}_{\mathrm{obs.}}|)}{\max\left(\log(1+|\tilde{u}_{\mathrm{obs.}}|)\right)}, \tag{2}$$

based on the phase shift ($\phi - \phi_{\mathrm{obs}}$) for a given time window between observed and synthetic data from their time-frequency representations (calculated using the Gabor transform). A logarithmic time-frequency weighting function $W_p$ determines the

regions within that space that have sufficient amplitude such that meaningful measurements can be made, with $\tilde{u}$ being the time-frequency representation of seismic signal $u$ as calculated via the Gabor transform. Misfits for all windows, traces, and events are summed to produce the total misfit for a given model.

The advantage of this misfit definition is that it combines the sensitivity to phase shifts from, for instance, a cross-correlation time shift misfit (Luo and Schuster, 1991; Dahlen et al., 2000) and to the shape of waveforms from an $L_2$-norm misfit (e.g.

Tarantola, 1984, 1986). It does not require the isolation of specific phases, and is therefore specifically suitable for interfering phases. However, it is beneficial to isolate small- and large-amplitude signals in separate windows such that the information from small-amplitude signals is not suppressed by the weighting $W_p$ (Section 4.6).





### 4.3 Optimisation algorithm

The objective functional in Equation 2 is minimised such that the synthetic seismograms for each consecutive model provide
a better match with the observed data than the previous. We use a conjugate gradient scheme (Nocedal and Wright, 2006;
Fletcher and Reeves, 1964) to compute model updates. This algorithm makes use of the gradient of the current iteration's
misfit with respect to the model parameters and a recursive term based on the previous iteration's descent direction.

The misfit gradient is constructed from sensitivity kernels obtained using the adjoint method (e.g. Tarantola, 1988; Tromp
et al., 2005; Fichtner et al., 2006; Fichtner, 2010). The raw gradients for each model parameter are preprocessed before a
descent direction is computed in order to improve convergence properties of the gradients. Kernels for each event are clipped
at the 99th percentile in order to avoid too-strong localisation of updates especially in the source region, and then summed to
produce the misfit gradient. The side and bottom edges are set to zero to remove potential boundary effects, and some smoothing
is applied. This processing routine is based on experience from previous inversions and some initial experimentation. It is re-
evaluated at several points during the inversion (Supplementary Table 2). The processed gradients are used to compute a descent
direction using the conjugate gradient scheme, and the step length is then determined using a quadratic interpolation between
the current model and three test models made with steps of lengths of 5%, 7% and 10% of the maximum gradient amplitude.
The final model for an iteration is obtained from the computed descent direction and the step length thus obtained.

### 4.4 Choice of initial model

Because of the local nature of the gradient-based optimisation, the choice of initial model is of crucial importance: the closer
the initial model is to the global minimiser, the more likely it is that it lies within the same "misfit valley".

Fichtner et al. (2013b) constructed a model of the European crust and upper mantle (with an embedded higher resolution
model of Anatolia (Fichtner et al., 2013a)). These models were constructed using waveform tomography and already include
considerable detail. The starting model for the current study is retrieved from these, by way of the initial stages of the Collab-
orative Seismic Earth Model (Afanasiev et al., 2015; Fichtner et al., 2018), which combines the local models with a smoother
background of S20RTS anomalies (Ritsema et al., 1999). The transition between these is smooth, owing to the efforts of
Afanasiev et al. (2015) to combine regional seismic tomography models into a coherent and consistent global model. This
gives a starting model with considerable detail in the northern parts of the model domain, but only very smooth structures in
the southern parts (Figure 2).

### 4.5 Multi-scale approach

To further mitigate the risk of descending towards (insignificant) local minima, we use a multi-scale approach (Bunks et al.,
1995). The lowest frequencies are inverted first, and as more of the data is explained by the model, higher frequencies are
included in a stepwise manner. As a result of this approach, the large-scale structure within the model domain is obtained
before small-scale details are filled in. This approach, a standard in waveform tomography (e.g. Akçelik et al., 2002; Tape
et al., 2007; Virieux and Operto, 2009), mitigates the risk of cycle skips, spares computational resources and increases (at





**Figure 2.** The starting model for (isotropic) S velocity, P velocity and density, as derived from Fichtner et al. (2013b, a). Slices are plotted at depths of 60, 100, 200 and 300 km, as relative deviations from the depth-averaged starting model. (a-d) isotropic S velocity $v_{\mathrm{S}}$, computed as the Voigt average of the anisotropic velocities $v_{\mathrm{SH}}$ and $v_{\mathrm{SV}}$ ($v_{\mathrm{S}} = \sqrt{\frac{1}{3}(v_{\mathrm{SH}}^2 + 2v_{\mathrm{SV}}^2)}$; e.g. Babuška and Cara, 1991), (e-h) P velocity $v_{\mathrm{P}}$, (i-l) density $\rho$. Note the different colour scales.



| # | Period range | Frequency band | Number of iterations | Simulated time | $n_x \cdot n_y \cdot n_z \cdot n_t$ |
|---|---|---|---|---|---|
| 0 | 100–150 s | 0.0067–0.01 Hz | 10 | 1200 s | $10.2 \times 10^6$ |
| 1 | 80–150 s | 0.0067–0.0125 Hz | 10 | 1200 s | $24.4 \times 10^6$ |
| 2 | 65–150 s | 0.0067–0.0154 Hz | 10 | 1200 s | $44.8 \times 10^6$ |
| 3 | 55–150 s | 0.0067–0.0182 Hz | 10 | 1001 s | $60.2 \times 10^6$ |
| 4 | 46–150 s | 0.0067–0.0217 Hz | 10 | 990 s | $89.1 \times 10^6$ |
| 5 | 38–150 s | 0.0067–0.0263 Hz | 15 | 990 s | $167.3 \times 10^6$ |
| 6 | 32–150 s | 0.0067–0.0313 Hz | 20 | 1000 s | $347.3 \times 10^6$ |
| 7 | 28–150 s | 0.0067–0.0357 Hz | 15 | 900 s | $570.2 \times 10^6$ |

**Table 1.** Overview of inversion choices. Within each frequency band, ∼10-20 iterations are carried out, starting with the longest periods (Bunks et al., 1995). The column 'Simulated time' shows the duration of each synthetic earthquake simulation. As frequency increases, the surface wave train becomes more compact (see Figure 3), so the simulation duration can be shortened. The final column $n_x \cdot n_y \cdot n_z \cdot n_t$ shows the product of the discretisation in the three spatial directions (number of elements) and time (number of time steps), which serves as an indication of the computational cost of a single forward simulation.

long periods) the proportion of the model domain that the data are sensitive to. This means that some of the long-wavelength structure of the southern, less well covered part of the model domain, can also be retrieved.

The used frequency bands are specified in Table 1. Care is taken to avoid the introduction of cycle skips when new frequency content is introduced in the inversion. This is done by limiting the highest frequency to be less than 1.25 times the previous highest frequency. In each frequency band, ∼10–20 iterations are carried out.

At each frequency band, the events to be included are re-evaluated, mainly for the benefit of reducing computational cost (see Table 1). At low frequencies, events with few stations may provide valuable constraints on parts of the model domain that are otherwise poorly covered – several such events are located in North Africa (Figure 1). These events are mostly discarded at higher frequencies, where the benefit of including the event becomes unfavourable compared to the computational cost.

In addition to the smoothing of the kernels as described in Section 4.3, we found it was necessary to apply additional
processing to the total model update at the end of the iterations in some of the frequency bands. This mainly involved additional smoothing and removal of edge effects. All additional smoothing and damping is described in Supplementary Table 2.

### 4.6 Waveform and window selection

For the initial three frequency bands (down to periods of 65 s), the first 1200 s of data after the event origin time were evaluated. This is reduced in subsequent period bands, where the surface wave train becomes more compact (see Figure 3) and the compu-
tational cost per simulation increases (Table 1). Ideally, complete seismograms are used as data in the inversion. However, due to potential source- and receiver-side issues, geometry and noise, some selection of the data needs to be performed (Krischer et al., 2015). The initial window selection is carried out automatically using LASIF. For the first three frequency bands, windows are reviewed manually for all 50 000 seismograms; windows for later frequency bands are reviewed in part. During some
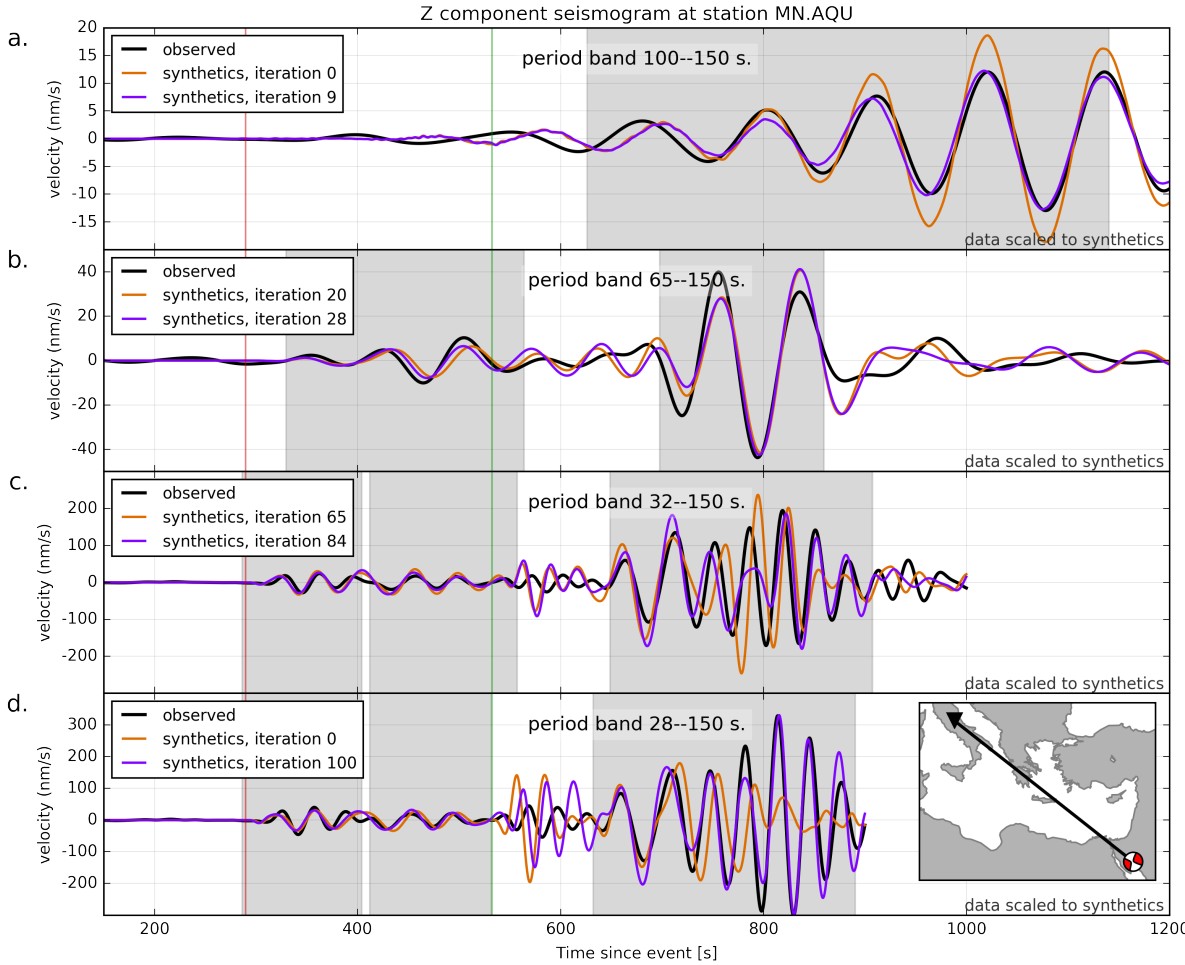

**Figure 3.** An illustration of the vertical-component seismogram recorded at MedNet station MN.AQU resulting from an earthquake in the Sinai region (27 June 2015, 15:34:03 UTC, $M_W$=5.6 – see inset) and corresponding synthetics for different period bands and iterations (see Table 1). Selected windows are shown as grey shaded areas. Vertical lines indicate P-wave (red) and S-wave (green) first arrival times predicted using TauP toolkit (Crotwell et al., 1999) in ObsPy (Beyreuther et al., 2010) for PREM (Dziewoński and Anderson, 1981). In the final period band (panel d), we compare the initial model 0 with the final model. The fit of the body wave data improves visibly, but especially the fit of the surface wave train improves dramatically – this observation is typical for the entire dataset.

initial experimentation, it was found that the inclusion of late arrivals could contribute to the formation of boundary artefacts,
especially if source or receiver are positioned close to the domain boundary. Such windows are therefore excluded.

Our window selection strategy is aimed to maximise sensitivity to deep structure. This is done by explicitly separating small- and large-amplitude signals into different windows, which allows us to make use of the (smaller amplitude) body wave information. Figure 4 illustrates this. Two separate windows A and B reveal significant and complementary information in both





wave packets, which is demonstrated by the deep sensitivity for $v_P$ for body wave window A. In the combined window A+B,
both the misfit and the $v_P$ sensitivity kernel are dominated by the large-amplitude surface-wave signal, which is not significantly
different from window B. Separating such windows accellerates the convergence of the body wave data and thus, given the
finite total number iterations, allows us to better resolve deeper structure.

## 5 Results

A total of 100 conjugate-gradient model updates were calculated, divided over eight consecutive frequency bands going down
to periods of 28 s (Table 1). In this section, we will discuss the resulting models and misfit development.

### 5.1 Misfit development

Figure 5 displays the misfit development for all the frequency bands used in this study. The misfit reduction varies per frequency
band but is on the order of 10–20% within each band.

Comparing the initial and final model in the period band of 28–150 s, the misfit drops by 48%. This is demonstrated in
Figures 6 and 7, where we compare observed traces for this period band with synthetics from the initial and final model
using windows from the final iteration. Figure 6 demonstrates that the most important misfit improvements come from events
clustering in the central Aegean part of the model domain, and towards the south-east. The events with the largest initial misfit
(which are the events with the largest number of windows) display the largest drop in misfit – both in absolute and in relative
terms. Phase shifts per window become closer to zero for the final model.

Figure 7 shows examples of seismogram fits for an event in the Aegean sea. Especially at stations farther away from the
source (e.g. stations 1 and 9), the difference between the synthetic seismograms for the initial and final models is striking.

### 5.2 Model

After 100 iterations, the model has been updated considerably for all parameters down to the transition zone (Figure 8).
Updates are strongest near the surface, and decrease in strength with depth. P-velocity is updated significantly less strongly
than the other parameters, and to shallower depths (Figure 8a). Average shear velocities do not change significantly from the
starting model (Figure 8b), with average $v_{SH}$ slightly higher than $v_{SV}$ in the uppermost 100 km. Figure 9 shows how the Hellenic
subduction zone becomes progressively more pronounced as iterations progress.

The final model is shown in Figure 10 (compare to the starting model in Figure 2). In general, the S velocity model has
features that can be linked most coherently with our knowledge of the geological situation. In the upper ∼100 km, the con-
tinental areas have slow velocities while the old oceanic lithosphere of the Central/Eastern Mediterranean is fast. In contrast
to this, both the Western Mediterranean and the Aegean sea display low velocities, as expected from their history of young
oceanic crust formation and recent or ongoing extension. Several high-velocity zones can be linked to areas of known current
or previous subduction. These will be discussed in more detail below.





**Figure 4.** An illustration of the difference between selecting multiple windows or a single window in the presence of both small-amplitude and large-amplitude signals. (a) Observed and synthetic seismograms from an earthquake in the Aegean (4 April 2014, 20:08:08 UTC, $M_W$=5.6), band-pass filtered between periods of 46 and 150 s, with two separate windows A and B. (b) Weighted time-frequency phase difference $W_p (\phi - \phi_{\text{obs.}})$ (Equation 2) between observed and synthetic seismograms for window A. (d) A cross-section through the corresponding sensitivity kernel for $v_P$. (c,e) Phase difference and kernel for window B. (f) The same traces as in (a), but now with a single combined window A+B. (g) Map showing the location of the cross-section, with the locations of the earthquake and station indicated by a red circle and yellow triangle, respectively. (h,i) Phase difference and kernel for the combined window A+B. Note the similarity to the corresponding plots for window B (c,e) – a result of the weighting $W_p$ that suppresses the effect of the small-amplitude signal.





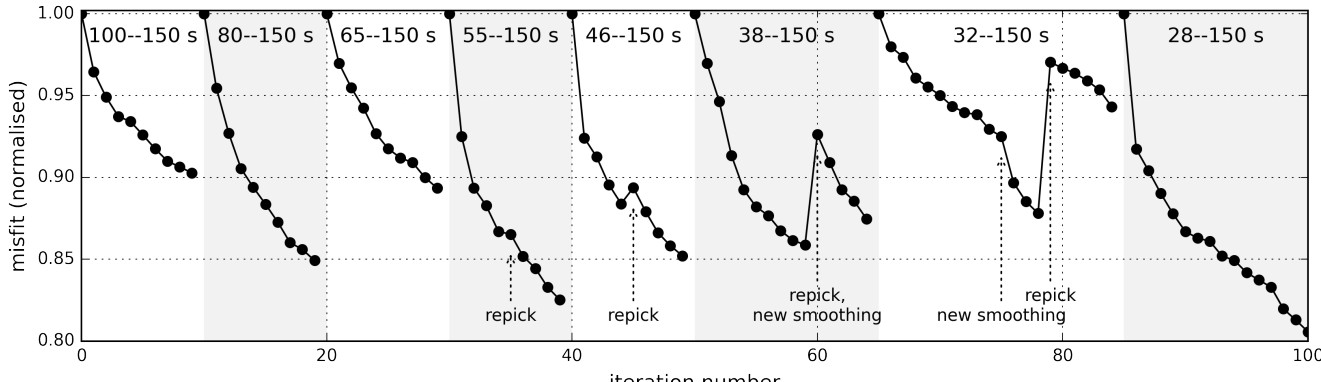

**Figure 5.** Misfit development across iterations, normalised by the initial misfit within each frequency band. Within every frequency band, 10–20 conjugate-gradient updates are calculated (Table 1). In the period bands of 55–150 s and higher, windows were re-evaluated mid-period band, indicated with the label 'repick'. The increase in misfit at these points is a result of the fact that the objective functional is changed as additional windows are included – a result of the fact that the waveforms grow more similar. In some cases, the smoothing parameters (Section 4.3) are also changed mid-period band. This is done when the misfit development starts flattening out.

The picture is not so clear for P velocity and density. P velocity has generally increased in the upper ∼200 km of the model
domain, with smaller amplitudes than in S velocity. Because for a given frequency, wavelengths are longer for P waves, the resulting model is smoother than the S velocity model. The Hellenic subduction zone is visible down to ∼250 km. In the density model, the opposite effect is visible: because it is mainly contrasts in density to which the waves are sensitive, the updated structures are of generally shorter wavelength. As a result, the imaged structures are of stronger amplitude and tend to be more oscillatory. Although this reaches greater depths (even beyond the transition zone), the results are much less coherent.
We will discuss this separately below.

**5.2.1 High-velocity structures**

Several distinct high-velocity structures are visible within the model. A 3-D rendering of these structures is shown in Figure 11, visualised with the 4.75 km/s velocity isosurface. This value is chosen because it is somewhat above the upper-mantle average, and serves to emphasise the approximate outline of the high-velocity features. Cross-sections are shown in Figures 12 and 13.
Beneath Italy, a high-velocity body (labelled 'A' in Figure 11) stretches along most of the Italian peninsula underneath the Appenines towards Puglia and Sicily, visible between depths of ∼200–500 km (Figure 12a,b). This structure, mostly clearly separated from the surface, is imaged in many other studies and is sometimes interpreted as the remains of Tethyan subduction or as delamination from the Italian peninsula (e.g. Piromallo and Morelli, 2003; Koulakov et al., 2015).

On the other side of the Adriatic sea, a high-velocity anomaly (labelled 'B' in Figure 11) stretches from the southern Dinar-
ides south towards northern Greece. It is especially prominent underneath the Southern Dinarides (Figure 12b) but stretches along most of the Adriatic coast (Figure 12c). In the north, an anomaly is only visible near the surface underneath the Adriatic



**Figure 6.** A comparison of the initial and final models in the shortest period band (28–150 s). (a) Geographical distribution of the absolute change in misfit for each of the events from the initial to the final model. (b) A comparison of maximum time-frequency phase shift between observed and synthetic seismograms within windows, for the initial (black) and final (green) models. A positive phase shift means that synthetics are ahead of observed data, a negative phase shift the other way around. (c) Misfit for the initial (grey) and final (green/red) model, plotted both for the whole dataset (thick bar) and per event (narrow bars). Event values are scaled by the largest initial misfit and sorted by initial misfit. Total misfit decrease in this period band is 48%. The strongest misfit decrease is generally seen for the events with the largest initial misfit.

**Figure 7.** Misfit change for a single event, and example seismograms. (a) change in misfit for all stations of an event in the Aegean Sea (8 January 2013, 14:16:11 UTC, $M_W$=5.8), evaluated in period band 7 (28–150 s) (Table 1). Each dot represents the total change in misfit for a station. (b-j) examples comparing observed (black) and synthetic seismograms for the indicated stations (initial model: dashed, pink; final model: red). Vertical lines indicate P-wave (red) and S-wave (green) arrival times predicted for PREM (Dziewoński and Anderson, 1981) using TauP toolkit (Crotwell et al., 1999) in ObsPy (Beyreuther et al., 2010).





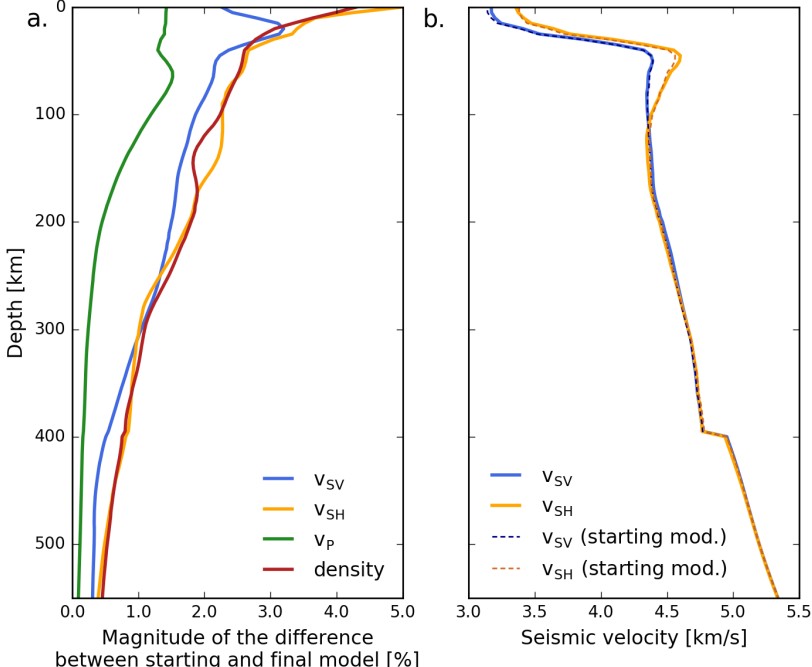

**Figure 8.** a. Depth average of the magnitude of the relative difference between initial and final model for all model parameters individually: $\mathrm{mean}(|(\mathbf{m}_f - \mathbf{m}_i)/\mathbf{m}_i|)$. b. The depth-averaged horizontal and vertical shear wave velocities for the initial and final models.

sea (Figure 12a). This Dinaric anomaly is imaged in other studies (e.g. Piromallo and Morelli, 2003) and correlates with the location of geodetically inferred convergence and subduction of Adria beneath Eurasia (e.g. Bennett et al., 2008).

The most prominent and strongest high-velocity structure sits beneath the Hellenic Arc and Aegean Sea (labelled 'C' in

Figure 11), and has been widely interpreted as the African plate subducting beneath Eurasia. At the surface, it follows the curvature of the Hellenic Arc from the Peloppones towards Crete. The anomaly dips downward and inward in a north-easterly direction down to the top of the mantle transition zone (Figure 13d,e). This structure has been imaged to various depths from the early days of seismic tomography (e.g. Spakman et al., 1988; Piromallo and Morelli, 1997; Amaru, 2007; Biryol et al., 2011; Portner et al., 2018; Hosseini, 2016). In our model, the fast anomaly flattens out towards the 410 km discontinuity and is

not imaged in deeper parts (Figure 13d,e).

Beneath western/central Anatolia, another high-velocity anomaly of large amplitude (labelled 'D') is visible, dipping northward from the Gulf of Antalya towards the transition zone (Figures 13f,g). It is imaged in various other studies (e.g. Biryol et al., 2011; Portner et al., 2018) and is interpreted as the deeper part of the Cyprean slab. In previous studies, the shallow part of this slab is imaged as near-horizontal, stretching northward from the Cyprus trench (e.g. Bakırcı et al., 2012). At the surface,

this part thought by some to be separated from the Hellenic slab via the Pliny-Strabo STEP fault (Govers and Wortel, 2005;





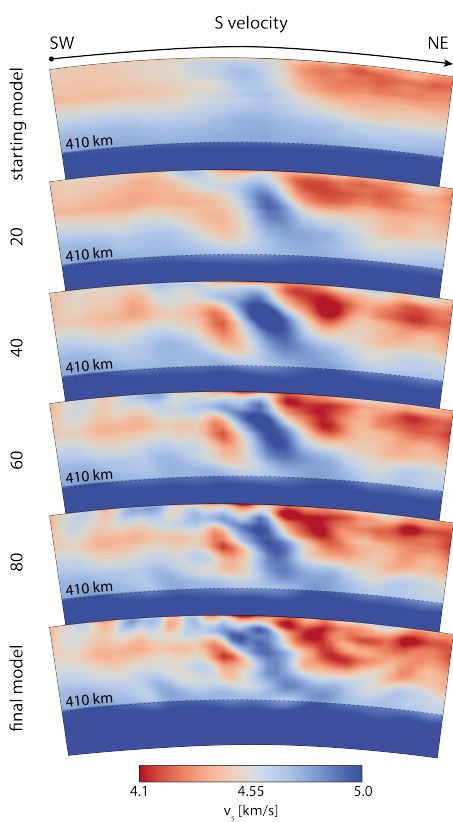

**Figure 9.** Development of the S velocity model across iterations, in a southwest-to-northeast section that is perpendicular to the Hellenic subduction zone (see Figure 1a). Plotted is the absolute isotropic S velocity $v_\mathrm{s}$. The starting model (Figure 2) is derived from Fichtner et al. (2013b). The section corresponds to the Hellenic section in Figure 13d (dark green).

Özbakır et al., 2013), whereas others surmise the Pliny and Strabo faults to be part of a general African-Eurasian convergence system (Howell et al., 2017). In our images, this shallow part is imaged intermittently and a clear gap is visible between the shallow and deep parts of the slab, which extends into the transition zone (Figure 13f,g).

A fifth anomaly, labelled 'E', is visible south of the Hellenic Arc in Figure 11. We will not interpret this further due to the
poor data coverage in this area, mostly derived from a single earthquake.




**Figure 10.** The final model for isotropic S velocity, P velocity and density – compare to Figure 2. Slices are plotted at depths of 60, 100, 200 and 300 km, as relative deviations from the depth-averaged starting model. (a-d) isotropic S velocity $v_S$, (e-h) P velocity $v_P$, (i-l) density. Note the different colour scales, where especially the amplitudes of variation in P velocity are much lower.



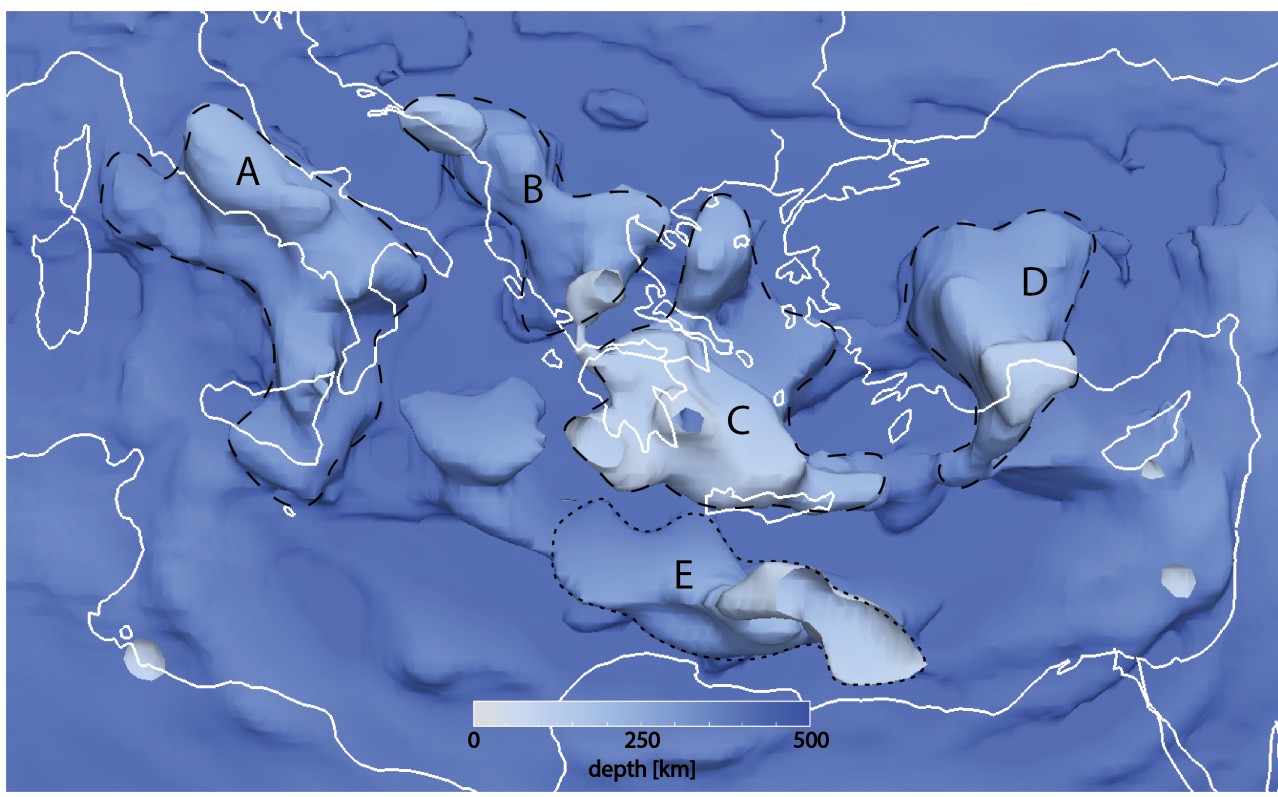

**Figure 11.** A 3-D rendering of high-velocity structures within the model domain. For this figure, the 4.75 km/s isosurface of isotropic S velocity $v_S$ was selected. This value is chosen because it is somewhat above the upper-mantle average, and serves to emphasise the approximate outline of the high-velocity features. The model is coloured by depth, with shallower regions whiter and deeper regions bluer. The high velocities are labeled with letters: A: Italy/Appenines; B: Dinarides, C: Hellenic, D: Anatolia. E is possibly an artefact. A 3-D visualisation video is available through the Supplementary material, as well as the vtk file for this figure – this can be viewed using e.g. ParaView (Ahrens et al., 2005).



**Figure 12.** Cross-sections through the S velocity model and a map showing their locations. Left column: isotropic S velocity $v_S$, right column: relative deviations in $v_S$ from the depth-averaged starting model. Black dots indicate seismicity in the region, as obtained from the EMSC-CSEM catalogue (2004–August 2019, depths greater than 40 km and M>2; Godey et al., 2013). Seismicity on the cross-sections is plotted if it is within 50 km from the cross-section slice. (top) Map showing the locations of the cross-sections. (a) Cross-section across Northern Italy (anomaly A) (b) Cross-section across Italy and the Dinarides (anomalies A & B) (c) Cross-section parallel to the Dinaric anomaly (anomaly B)





**Figure 13.** Continuation of cross-sections from Figure 12 (d) Cross-section across the Hellenic subduction zone (anomaly C) (e) Alternate orientation cross-section of the Hellenic subduction zone (anomaly C) (f) Cross-section across the Anatolian subduction zone (anomaly D). (g) Alternate cross-section across the Anatolian subduction zone.




## 6   Discussion

### 6.1   Advantages and limitations of the method and model

Although computationally more expensive than ray-based tomography methods, the advantage of waveform tomography lies in its ability to make use of all and any part of seismograms that has a sufficient signal-to-noise ratio, allowing for a more complete
extraction of all information that is carried by seismic waves. This allows us to take into account body and multimode surface waves, source effects, and frequency-dependence, and jointly invert for multiple parameters for the crust and upper mantle. This is done in a manner where the misfit is computed directly from the observed and simulated seismograms, thus providing a self-consistent inversion framework. This method is particularly powerful in tectonically active parts of the Earth such as the Mediterranean, where strong heterogeneity is unavoidably present – circumstances under which the assumptions underlying
ray theory become invalid.

Nevertheless, it is important to understand that the results are unavoidably affected by observational and methodological errors. In the following paragraphs we will discuss some of these issues.

Moment tensors used in this inversion are taken from the Global Centroid Moment Tensor catalog (Ekström et al., 2012) using the point source approximation. These are not inverted for separately, as limited azimuthal coverage – in particular near
the model domain boundaries – may lead to a bias in the results, as well as potential overfitting. Valentine and Woodhouse (2010) note that the model used for source inversions can have an imprint on tomographic results. Particularly in areas of limited coverage errors in the description of the source can have a large effect – this may be the case for Anomaly E in Figure 11. In order to minimise such effects, patterns in phase shift for each event are monitored manually and in some cases, source parameters were adjusted accordingly. In line with results from Bozdağ et al. (2016) and Hjörleifsdóttir and Ekström (2010),
these adjustments are generally on the order of a few s or km, and they correspond more closely to the reported ISC-EHB locations (Engdahl et al., 1998; Weston et al., 2018) as well as results reported in regional tectonic studies such as Howell et al. (2017). Examples are shown in Section 2 of the Supplementary material. In a few cases, events were removed from the inversion altogether, as their waveforms showed a notable jump in complexity in the higher frequency bands which may point to issues with the point source assumption or the source time function.

The wave propagation simulations are carried out on a regular mesh: a spherical chunk that includes no ellipticity, topography, ocean layer, or explicitly meshed internal discontinuities. The neglect of topography can lead to small-scale artefacts near the surface, in particular if the topography is on the same length scale as the minimum wavelength (Nuber et al., 2016). Similarly, internal discontinuities and the effect of the water layer are not modelled explicitly. Especially near the surface, the model thus has to be interpreted as an effective representation of the real Earth, valid for the frequencies used in the inversion
(Capdeville et al., 2010). Sharp or small-scale layering and (apparent) anisotropy thus cannot be distinguished reliably.

Another source of error are the actual data used in the inversion. Such errors can pertain to the location, orientation and timing of the receivers, as well as instrument errors and errors in the reported response. Many of these issues are caught using the automated quality control carried out in our workflow, but subtle errors remain difficult to catch. In areas of dense data coverage, this will average out, but in the regions of poor coverage, these may still affect the inverted result.





## 6.2 Fit to data not used in the inversion

Within the framework of the inversion and issues as outlined above, it is possible to analyse the trustworthiness (and limitations) of the model.

For this, we compute the misfit for the initial and final models for six earthquakes that were not part of the inversion (Supplementary Table 1). We compare complete traces where no window selection has taken place in order to avoid preferential usage of data that 'work well'. For this reason, and following Tape et al. (2010) and Simutė et al. (2016), we use for this a normalised $L_2$ waveform difference misfit:

$$J_{L_2} = \frac{\int_t \left[\mathbf{u}(\mathbf{m}, t) - \mathbf{u}_{\text{obs.}}(t)\right]^2 \mathrm{d}t}{\sqrt{\int_t \left[\mathbf{u}(\mathbf{m}, t)\right]^2 \mathrm{d}t \int_t \left[\mathbf{u}_{\text{obs.}}(t)\right]^2 \mathrm{d}t}}. \tag{3}$$

Figure 14 shows the change in misfit for these events for all stations as well as example seismograms from events A–E (Supplementary Table 1). The data fit improves both visually and quantitatively in the large majority of traces.

## 6.3 Spike tests

In addition, we assess our model with what can be viewed as a "waveform tomography equivalent" for spike testing. While more sophisticated methods, such as resolution analysis through random probing (Fichtner and van Leeuwen, 2015), do exist, spike tests do not rely on any assumptions about the shape of point spread functions which – especially in the case of density – may be strongly non-Gaussian. They thus give qualitative insight into smearing in a way that is visually easy to interpret.

As Rawlinson and Spakman (2016) note, it is important in ray-theory traveltime tomography to use the raypaths of the actual model to assess recovery, smearing and trade-offs. To mimic this using the full wavefields computed from 3-D wave propagation, we add a Gaussian anomaly onto one parameter in model 91 and use the real data to run five additional iterations in the final period band of 28–150 s. If the model is robust, the data will detect the (presumably) erroneous spike, and updates will be generated that remove the anomaly. We do this for each parameter ($v_{\text{SV}}$, $v_{\text{SH}}$, $v_{\text{P}}$ and density) individually, thereby also checking trade-offs. Spikes are introduced in an alternating chequered pattern at different depth levels.

In Figure 15, we compare, at a depth of 100 km, the input anomaly (top row) with the part of the model update that is a result of the addition of the spikes (bottom four rows). This is computed by subtracting the reference model update (without spikes) from the update after five iterations of the spiked model: $\mathbf{u}_{\text{spike}} = (\mathbf{m}_{s+5} - \mathbf{m}_s) - (\mathbf{m}_{96} - \mathbf{m}_{91})$.

Comparing Figure 15 to Figure 1, we immediately observe that those parts of the model domain that are well-covered are indeed also the parts in which the spikes are most successfully removed. This is most visible for the first two rows, with spikes in $v_{\text{SH}}$ and $v_{\text{SV}}$, respectively. These are also the parameters that are recovered best, with trade-offs to other parameters that are less significant. Recovery of $v_{\text{P}}$ is, in general, very limited. Trade-offs from $v_{\text{P}}$ to other parameters are partially stronger than the updates in $v_{\text{P}}$ itself. This is in line with our observations of the final model (Section 5.2 and Figure 10).

Recovery of density is also limited. Only in the central Aegean recovery is of a level approaching that of $v_{\text{SV}}$ and $v_{\text{SH}}$, and the negative spikes are surrounded by a positive "halo". Furthermore, when comparing the different rows, we see that trade-offs *to* density from the other parameters are larger than the updates in density itself. While this is a test done over only a few





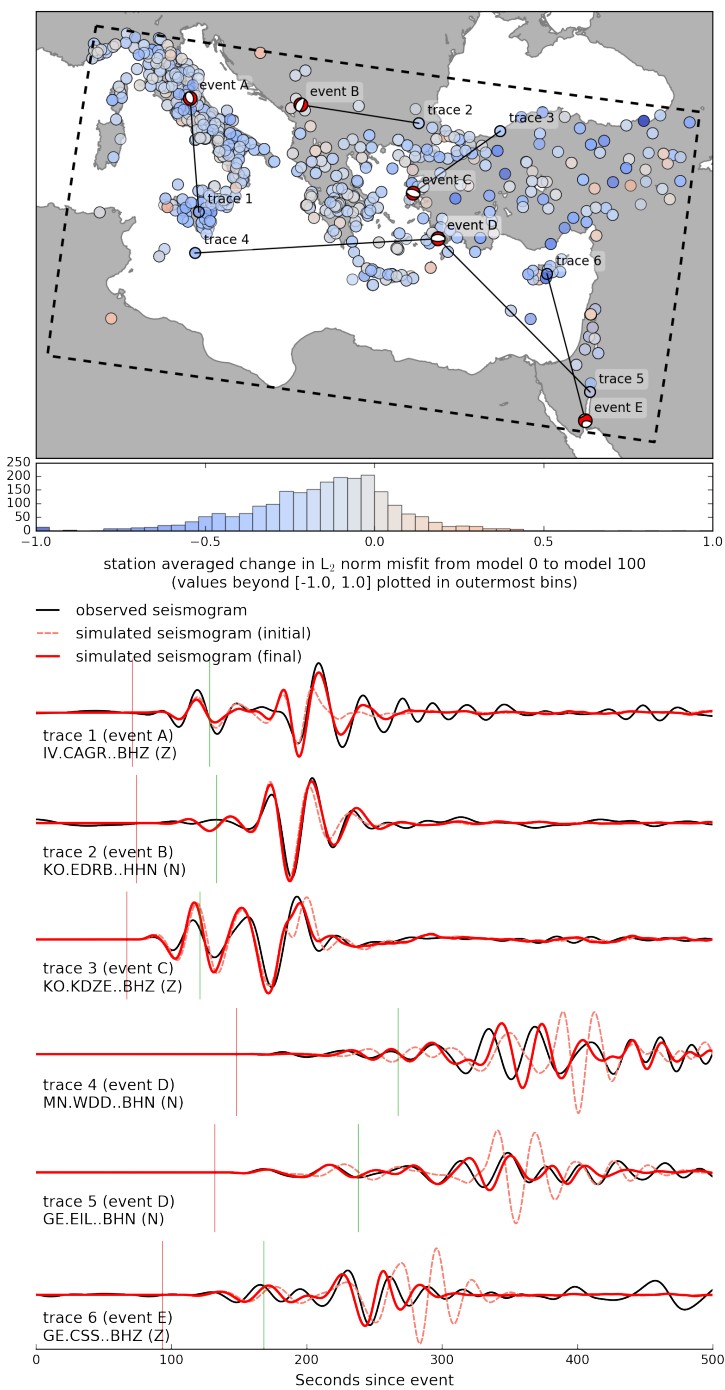

**Figure 14.** $L_2$ norm misfit decrease for new events and example traces. Top: Map with for each station a dot representing the station-averaged change in $L_2$ norm misfit for whole traces for all new events, and histogram with the same data. Misfit decreases for 79% of the data. Total $L_2$ norm misfit decreases by 9%, compared to 13% for the original data set. Bottom: Example seismograms showing observed data (black) compared with synthetics for the initial (dashed, pink) and final (red) models for the indicated source-receiver pairs.




iterations and it is possible that these trade-offs even out more as iterations progress, this does indicate that density is not a stable parameter.

In previous experiments, we showed that density has a clear imprint on the seismic wavefield and can in principle be inverted
for (Płonka et al., 2016; Blom et al., 2017). This is especially the case at longer periods (Takeuchi and Saito, 1972). The fact that density cannot be imaged reliably in the present study can be attributed to two main factors. Firstly, as density affects the seismic wavefield mainly through reflection/back-scattering, it is predominantly *contrasts* in density that are imaged. This results in a sensitivity to density at much shorter lenght-scales than for S velocity. It is therefore likely that short-scale structure from other parameters is mapped into density – this is especially likely near sharp transitions such as the surface and the
Mohorovičić discontinuity, and may be exacerbated by the lack of topography/bathymetry and the ocean layer in our mesh. Furthermore, variations in density have a smaller effect on the seismic wavefield than variations in seismic velocity (Blom et al., 2017). This means that in the presence of noise, inaccuracies in the modelling method as discussed above, or variations in parameters not included explicitly in the inversion (e.g. attenuation or further anisotropy), density is likely to act as an "inversion garbage bin". For this reason, we presently do not further interpret density. A more accurate treatment of the effects
named above is the focus of a future study.

## 7   Conclusions

We have imaged the upper mantle beneath the Central and Eastern Mediterranean using waveform tomography, simultaneously inverting for radially anisotropic S velocity, P velocity and density. We particularly aimed to resolve deep structure, by using an approach that combines the use of long period data at the beginning of the inversion with a windowing technique that optimally
makes use of the separation of small-amplitude body waves from large-amplitude surface waves. This has resulted in a model in which several high-velocity structures are imaged down to the transition zone, which can be correlated to the current and past tectonic setting. This model is able to explain new data not used in the inversion.

Due to the natural dominance of surface wave signals in waveform tomography, our model is best constrained for S velocity, and we therefore base our interpretations on this parameter. P-wave velocity structure is less well recovered and smoother
– a result of its longer wavelength for the same frequency – but broadly speaking it displays the same structures. Density, meanwhile, is poorly constrained, due to the sensitivity of the measurements to contrasts in density rather than the parameter itself, and due to the fact that of the imaged quantities, it has the smallest imprint on the wavefield, making it more sensitive to data or modelling errors. These observations are demonstrated using spike tests, and efforts are ongoing to improve the joint imaging of all parameters.
The Hellenic slab is the most prominent feature in the model. It extends from the surface down to the transition zone in a bent, arcuate shape. It is visible in both S and P velocity. Two more or less parallel anomalies are visible beneath the Italian Peninsula and the Dinarides. The Italian anomaly is visible at depths below ∼100 km, whereas the Dinaric anomaly extends towards the surface beneath the southern Dinarides. Although extending towards Greece in the south, the Dinaridic anomaly is well separated from the Hellenic slab. The final large feature in the model is the high-velocity anomaly beneath Anatolia,



**Figure 15.** Spike tests conducted for model 91 in the period band 28–150 s, designed in the spirit of Rawlinson and Spakman (2016). Each of the columns represents a separate test with a checquered grid of spikes in one of the parameters. We denote the original model as $\mathbf{m}_{91}$, the model with spikes as $\mathbf{m}_s$ and the model produced after 5 iterations as $\mathbf{m}_{s+5}$. The updated reference model is $\mathbf{m}_{96}$. The top row represents the input anomaly for that particular parameter, where we show $\mathbf{m}_o - \mathbf{m}_s$. In the following four rows, we show the cumulative model update after five iterations – i.e. $(\mathbf{m}_{s+5} - \mathbf{m}_s) - (\mathbf{m}_{96} - \mathbf{m}_{91})$ – for each of the parameters. Results are shown for a depth of 100 km.



interpreted as the Cyprus slab. This structure dips towards the north and east beneath the western and central parts of Anatolia. No clear connection between the shallow and dipping parts of the anomaly are imaged, and it is also well separated from the Hellenic slab.

*Code and data availability.* All data were downloaded from publicly available data repositories such as IRIS (www.iris.edu) and ORFEUS (www.orfeus-eu.org) using the ObsPy/LASIF mass downloader. Processing and workflow took place using LASIF (Krischer et al., 2015,
downloadable via http://lasif.net) and ObsPy (Beyreuther et al., 2010, https://www.obspy.org), which are free open source Python libraries. The simulations themselves were run using the wave propagation package SES3D. This is free open source software released under Apache 2.0 License. It is available for download at https://cos.ethz.ch/software/production/ses3d.html. The final model is available as ascii and vtk files, the former suitable for interaction with ses3d, the latter suitable for viewing with ParaView.

*Video supplement.* A video demonstration of the final model is available through the Supplementary material.

*Author contributions.* NB mainly designed the project, conducted the research and wrote most of the manuscript, in collaboration with AF. AG designed the optimisation algorithm and wrote the code that interfaces the simulations and inversion workflow, specifically adapted to the research project and to the cluster that the simulations were run on. AF helped design the project and supervised the execution, and helped write the manuscript.

*Competing interests.* No competing interest are present.

*Acknowledgements.* This research was funded by Netherlands Organisation for Scientific Research NWO grant under Project Number 864.11.008, the Swiss National Supercomputing Center (CSCS) in the form of the ch1 and s868 projects and the European Research Council (ERC) under the European Union's Horizon 2020 research and innovation programme under grant agreement No 714069. We furthermore gratefully acknowledge the enormous quantities of freely available seismic data that are continuously collected by and made available to the seismological community. IRIS and ORFEUS, as well as institutional data centres play no small role in facilitating this. LASIF was the
main workhorse for carrying out all the data and window selection in this project. We thank Lion Krischer, who wrote the original version used in this project, and all others who have contributed to this freely available tool. This would not be possible without the seismological functionality that has been developed in the ObsPy toolkit. All simulations made use of ses3d and SISYPHUS, which were run on the CSCS Swiss National Supercomputing Centre cluster Piz Daint. We are grateful to all CSCS staff who were always quick to help out and get things working. Finally, we thank Rob Govers and Deborah Wehner for carefully reading the manuscript with the critical eye of the outsider.





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
