# Peer review of "Seismic waveform tomography of the Central and Eastern Mediterranean upper mantle"

_Solid Earth, 2019_

## Referee Comment (RC1) · Anonymous Referee #1 · 25 Dec 2019

In this manuscript, a new seismic model for the Central and Eastern Mediterranean is constructed based on full-waveform inversion (FWI). The authors not only inverted for vertically and horizontally shear and P wavespeeds but also density in their iterative model updates — only the phase information used in the measurements, excluding amplitudes. The main focus is on the methodological part where the detailed model interpretation is reserved for a potential future paper.

It is generally a well-written manuscript, however, there are a few issues to be highlighted. The authors state that the focus is on the methodological part which is pretty similar to Fichtner et al. and Krisher et al. except for inverting for density. The method

part may be shorten to avoid repetitions. The main difference compared to most of the previous FWI studies is the simultaneous inversion of density with wavespeeds. However, there are also other examples in literature from exploration to lithospheric studies where density is also inverted, but they are not cited. It is worthwhile to have a complete literature review, and it would also be more insightful to compare the results of density inversions to those previously published tomographic studies.

On the other hand, the message of the paper needs a bit of clarification. The authors very briefly discuss the potential reasons for the failure of retrieving density in conclusions. However, it is not clear if density can indeed be retrieved or not in FWI. If not, is it better not to invert it, or is there any advantage of inverting for it despite the trade-off between seismic parameters? It would be helpful tp extend these discussions considering the results of other studies.

Some further detailed comments are as given below:   Page 2, line 40, last sentence: The authors state that not only P and S wavespeeds but also density can be constrained by full-waveform inversion. However, the results of the study seem not promising to support this statement. The authors discuss the potential reasons, but they do not discuss much what should be done to retrieve density or what the strategy in future studies should be. Will taking full anisotropy and attenuation into account in inversions help retrieve the density model correctly?Âǎ

Page 5, line 115: Parameterization is an important decision in FWI to capture the physics of the medium. It is not clear if the authors are inverting for only SV and SH wavespeeds for radial anisotropy or also the anisotropic parameter \eta? If the former, why not including \eta to more robustly consider the radial anisotropy?

Why was not the amplitude information used in the misfit, which may be more sensitive to density variations?

The manuscript nicely summarizes the strategies considered in the inversion algorithm. However, it is not described how the crust was considered in simulations and

inversions, which can have a significant effect on the constructed model. Besides, focusing on a subdomain of the model of Fichtner et al. by performing ~95 iterations, one would expect higher resolution in the model, which also deserves investigation of depths shallower than 60 km where some interesting crustal features may be observed.

Section 4.3: It would be good to mention the smoothing and/or pre-conditioning strategies if applied.Âă

Section 4.4: To my understanding, the starting model has multi-resolution (smooth global model, inverted European model, and higher-resolution Anatolian model), and a smoothed version of it is used. What is the resolution or the degree of smoothness of the chosen model to start iterations?

Figure 2: There are quite some small-scale variations (smaller than those in velocity models) in the starting model of density. How was density constructed in the starting model? How good is it to start with?Âă

Figure 6: Why is the histogram split into two parts? Why is there a gap around the zero phase shift?

Figure 10: Looks like there is a strong anti-correlation between P- and S-wavespeed models (i.e., the S-wavespeed model shows all slow wavespeed in continents, and fast wavespeed in oceans at shallower depths whereas the P-wavespeed model is predominantly fast). There is also a sharp boundary in the P-wave model on the Eastern part. How do the authors interpret these anomalies? Are these features also observed similarly in other tomographic models?Âă

Figure 15: Looks like the Gaussian anomalies used in the spike-tests of the P-wave model are larger than those of S-wavespeeds and density. Is it a plotting issue (or illusion?) or any specific reason for choosing it to be like that? It would be more insightful to show the spike-test results at other depths as well.

The authors have performed ~95 iterations, plus five additional iterations for every

Interactive
comment
parameter during the spike tests. It is a large number of iterations. How expensive is each iteration? Or what is the overall computational cost?Âă

---

## Referee Comment (RC2) · Anonymous Referee #2 · 18 Jan 2020

This paper presents a new seismic waveform tomographic model for the upper mantle beneath the Central and Eastern Mediterranean down to the mantle transition zone. They utilize information from body and multimode surface waves, source effects, anisotropy and attenuation. This manuscript mainly describes technical parts of this work, and does not provide detailed interpretation about their solutions. I do have a range of major concerns about the innovation and improvements of this work in comparison to previous work done by the same group, as well as other groups working on full waveform inversion and its applications for imaging the crust and upper mantle structures underneath Europe.

[Figure]

Major concerns:

(1) The authors claim that they also include radial anisotropy in their inversion, however, I cannot find any results for radial anisotropy in this manuscript. I strongly recommend the authors to present these solutions in order to make the statement consistent.

(2) It seems to me that the ETH group has published a number of papers on the same region (Europe) using the same approach (FWI). Why publishing another one or performing another inversion for the same target? What is the long term goal of this group for imaging the European upper mantle using FWI? Just including more data and publishing another paper once a while is not a good scientific exploration as far as I consider. For instance, I can list some papers which are similar to this manuscript. Can the authors provide some statements about significant improvements of this work in comparison to their previous work? Such as (1) Fichtner and Villasenor, 2015, EPSL; (2) Rickers et al, 2013, EPSL; (3) Fichtner et al, 2013, EPSL.

(3) In fact, From Figures 7 and 14, I am surprised to see that predicted seismograms from their starting model (Fichtner et al, 2013) have such large mismatches in comparison to observed waveforms, even for this long-period frequency band. If their previous model is already good enough, we should not see such significant differences.

(4) The authors just present their own tomographic model. In fact, the Mediterranean region is well studied by using either travel time tomography or even full waveform inversion by other groups. The authors ignored the same endeavor towards FWI imaging for the European crust and upper mantle by other groups. I wonder how their current model compares with other previous tomographic or FWI models? As far as I can tell from their Figure 12, the resolution for subducting slabs underneath the Mediterranean is quite low in comparison to previous travel time tomography (Wortel and Spakman, 2000, Piromallo and Morelli, 2003) or FWI models (Zhu et al, a series of studies). I strongly suggest the authors to make some comparisons with these previous studies, in order to convince the readers about the advantages or disadvantages of their current

inversion.

(5) The authors claim that their model has good resolution down to the mantle transition zone. However, in their horizontal cross sections (Figure 10), I only see results down to 300 km. I suggest they to put more slices, say from 400 to 700 km.

(6) For vertical cross sections (Figures 12 to 13). I don't see any reasons to put slices with absolute velocity here. Since they will be dominated by 1D structures, in particular for the mantle transition zone, which involves velocity jumps across 410 and 660. I suggest only show relative velocity perturbations in their right panel.

(7) In fact, for the Hellenic subduction (Figure 13), numerous previous studies have demonstrated that the Hellenic slab can penetrate the 660 km discontinuity down to the lower mantle (from either travel time tomography or FWI), however, the authors' result seems different from previous conclusions. Do we need to re-evaluate this consensus, or the authors' model does not have enough resolution to resolve it?

(8) For waveform comparisons, such as Figures 7 and 14, the authors only show comparisons between observed and predicted seismograms for a couple of station-source pairs. However, as pointed out by the authors, there are numerous seismograms. I am not surprised to see some improvements for some selected pairs of earthquakes and stations. How about other pairs? Since there are many stations in the authors' dataset, why not plot all three components (vertical, radial and transverse) for most data and predictions, just like common-shot gathers as did by exploration seismologists. I think that will be a better way to convince the readers that their model can really explain data, instead of just intensionally or randomly selecting some paris with good match. In addition, I suggest the authors to change color for predicted seismograms from the starting model, such as blue or other colors different from the current waveforms.

(9) Figure 3, why changing iteration numbers for different frequency bands? In order to demonstrate that their model can fit data for different frequency ranges, I suggest just compare the starting one with their current model (100th iteration).

(10) Figure 4, it seems to me that the color of panel d is highly saturated, then they can see a strong banana kernel, how about using some reasonable colors. In addition, do they need to add additional weighting coefficients in order to use both body and surface waves for this inversion?

(11) Figure 11, I suggest to plot iso-surface for relative velocity perturbation instead of absolute values, which is a typical way to examine subducting slabs.

(12) In section 6.2, in order to evaluate their model, the authors chose additional 6 earthquakes as an independent dataset. As far as I can tell, this is a very small dataset to convince the readers that their model can explain waveforms that are not used in the inversion. I strongly suggest them to increase the size of this independent dataset.

Some minor modifications:

(1) Section 1.1, the authors mentioned that the deeper parts of the model are mainly constrained by long-period data, I suggest also add body wave here. Since they provide more constraints for deep structures, in comparison to surface waves.

(2) Page 2, line 49, "manuscript"-> "paper"

(3) Page 3, line 62, what are these plates and micro-plates?

(4) Page 3, line 68, "trench roll back", I think it should be either trench retreat or slab roll back.

(5) Page 3, line 76, I don't understand the sentence starts with "West of this, . . ." I suggest to rewrite this sentence.

(6) Page 3, line 80, what is "STEP"?

(7) In section 3.2, I suggest the authors to add the numbers of stations as well as the numbers of measurements they have to constrain the current model.

(8) Page 5, line 105, "download" -> "observed"

(9) Section 4.3, line 143, "some smoothing", this is not clear to me, what is the smoothing and how large of it?

(10) Table 1, I think either use Period or Frequency, there is no need to use both of them.

My suggestion is major revision.

---

## Author Comment (AC1) · 21 Feb 2020

**Response to Reviewer 1**

Comments by Reviewer 1 in blue, answers in black.

It is generally a well-written manuscript, however, there are a few issues to be highlighted. The authors state that the focus is on the methodological part which is pretty similar to Fichtner et al. and Krisher et al. except for inverting for density. The method part may be shorten to avoid repetitions. The main difference compared to most of the previous FWI studies is the simultaneous inversion of density with wavespeeds. However, there are also other examples in literature from exploration to lithospheric studies where density is also inverted, but they are not cited. It is worthwhile to have a complete literature review, and it would also be more insightful to compare the results of density inversions to those previously published tomographic studies.

We feel that shortening the methodological part would risk deteriorating the repeatability of the work, and therefore respectfully prefer to keep it in the current format. While we have attempted to invert for density, we did not overly stress this particular aspect, because of the inconclusive results regarding this parameter in our study. Nevertheless, we have expanded the section discussing the density model and its limitations, referencing some relevant literature and discussing possible avenues to move ahead this field. This is now discussed in a separate section (Section 6.4).

On the other hand, the message of the paper needs a bit of clarification. The authors very briefly discuss the potential reasons for the failure of retrieving density in conclusions. However, it is not clear if density can indeed be retrieved or not in FWI. If not, is it better not to invert it, or is there any advantage of inverting for it despite the trade-off between seismic parameters? It would be helpful tp extend these discussions considering the results of other studies.

Especially in the case of exploration seismology, results regarding density seem to be encouraging. This has all to do with the fact that information derived from impedance contrasts can be used much more efficiently thanks to the availability of reflection data. Other factors include the controlled-source aspect of such studies and the source and receiver density. In favourable set-ups, such results may be used on lithospheric scales as well – especially if higher frequency data is used. In our case, however, the data are clearly transmission-dominated, meaning that reflection data are virtually absent. This is very similar in other applications at this spatial scale. Nevertheless, as we have shown in Blom et al. (2017), including density as a free parameter is important to avoid contamination of the reconstructed velocity distributions. A more in-depth discussion of density is included in Section 6.4 of the paper.

Some further detailed comments are as given below:

Page 2, line 40, last sentence: The authors state that not only P and S wavespeeds but also density can be constrained by full-waveform inversion. However, the results of the study seem not promising to support this statement. The authors discuss the potential reasons, but they do not discuss much what should be done to retrieve density or what the strategy in future studies should be. Will taking full anisotropy and attenuation into account in inversions help retrieve the density model correctly?

We agree that this was a deficiency of the original manuscript. This is improved in the revised

version. Though we were optimistic at the beginning of this study, we now see our ability to constrain 3-D density much more critically. Taking other parameters, including more complex anisotropy or attenuation, into account is unlikely to help. This is because any increase in the total number of model parameters can only act to decrease resolution. The actual crux of the problem is that sensitivity to density is so small that the recovered density model is strongly affected by unavoidably subjective regularisation. In this sense, a way forward would be to replace the current deterministic inversion by a fully probabilistic inversion that does not require regularisation. Then, at least, one would obtain complete and honest information about our state of knowledge on 3-D density inside the Earth. Research in this direction is underway [e.g., Gebraad et al., 2020. "Bayesian elastic full-waveform inversion using Hamiltonian Monte Carlo", JGR, in press]

Page 5, line 115: Parameterization is an important decision in FWI to capture the physics of the medium. It is not clear if the authors are inverting for only SV and SH wavespeeds for radial anisotropy or also the anisotropic parameter \eta.

The anisotropic parameter \eta is not inverted for. The more parameters are included in an inversion, the less well constrained it becomes. \eta is constrained by P-waves with variable incidence angle, for which we have too little data to constrain it properly. More broadly speaking, there is currently no 3-D heterogeneous model of \eta at regional to global scales. (This is different in exploration, where the necessary data may indeed be available.) We have formulated more clearly in the paper which exact parameters are inverted for (Section 4.1).

Why was not the amplitude information used in the misfit, which may be more sensitive to density variations?

Amplitudes are so sensitive to local velocity variations and source magnitude, that the risk of accidentally polluting the inversion would be too strong. Some amplitude information is used in the misfit indirectly: the relative amplitudes within a window. This does give valuable additional constraints on all parameters.

The manuscript nicely summarizes the strategies considered in the inversion algorithm. However, it is not described how the crust was considered in simulations and inversions, which can have a significant effect on the constructed model. Besides, focusing on a subdomain of the model of Fichtner et al. by performing ~95 iterations, one would expect higher resolution in the model, which also deserves investigation of depths shallower than 60 km where some interesting crustal features may be observed.

The initial crust was implemented in the form of a velocity gradient, meaning that the initial model has no sharp crust-mantle discontinuity. This approach is widely used in regional- to global-scale tomography [e.g., Fichtner & Igel, 2008; French et al., 2014], and it essentially served two purposes: (1) The absence of a discontinuity facilitates the construction of a finite-element mesh, and in particular, it avoids the presence of very small elements. (2) An initially smooth transition between crust and mantle allows the data to actually modify its sharpness. Thus, in the final model, the velocity gradient from crust to mantle is as sharp as required by the data. The disadvantage of this approach lies in a limited interpretability of the final model in the sense that no sharp separation between crust and mantle exists. (To illustrate resolution at shallower depth, we added to the Supplementary material a new depth slice at 50 km depth for

the spike tests.) The shortest periods in the simulations are 28 seconds, which limits the amount of detail present in the final model – irrespective of the number of iterations.

Section 4.3: It would be good to mention the smoothing and/or pre-conditioning strategies if applied.

In summary, the preconditioning amounts to the 99th percentile clipping, removal of information close to the edges, and smoothing. The latter two are informed by the frequency band (and resulting typical wavelengths) of the wave propagation in each iteration, and become smaller / finer as the inversion goes to shorter periods. The exact parameters used for each iteration are explained in full detail in the Supplementary Material, and we have added a clarification in the manuscript text referring to this.

Section 4.4: To my understanding, the starting model has multi-resolution (smooth global model, inverted European model, and higher-resolution Anatolian model), and a smoothed version of it is used. What is the resolution or the degree of smoothness of the chosen model to start iterations?

Indeed the starting model has several levels of resolution. However, no additional smoothing was applied prior to the inversion presented in this work. We have added information on the resolution lengths of the starting model.

Figure 2: There are quite some small-scale variations (smaller than those in velocity models) in the starting model of density. How was density constructed in the starting model? How good is it to start with?

In the starting model, density was not treated in any specific way, although it was updated along with the other parameters and also smoothed strongly. Because of this, density in the starting model has no particular interpretative value.

Figure 6: Why is the histogram split into two parts? Why is there a gap around the zero phase shift?

This is a result of the way the adjoint source computation is implemented for the time-frequency misfit. Internally, a decision algorithm attempts to filter out the windows in which the traces are too dissimilar to allow for a meaningful comparison. This algorithm happens to include a criterion based on a division by the maximum absolute phase shift within a TF window. This would be a very small number if the traces are very similar, and the resulting criterion big, which results in rejection. In other words, the window has a high risk of being (erroneously) rejected if the traces are very similar. This is of course a very crude algorithm, but the effect of this can be offset by slightly adapting the windows. As a result, however, the windows with a near-zero phase shift disappear from the distribution which is, as a result, very much non-Gaussian.

Figure 10: Looks like there is a strong anti-correlation between P- and S-wavespeed models (i.e., the S-wavespeed model shows all slow wavespeed in continents, and fast wavespeed in oceans at shallower depths whereas the P-wavespeed model is predominantly fast). There is also a sharp boundary in the P-wave model on the Eastern part. How do the authors interpret these anomalies? Are these features also observed similarly in other tomographic models?

We agree with the reviewer that at first sight there does appear to be an anticorrelation, however upon closer inspection this turns out to be not actually the case – the P-wave model is fast in

general. The sharp boundary in the P-wave model shows the boundary between the non-updated and updated parts of the model domain, i.e. where there is data coverage and where there is none.

Figure 15: Looks like the Gaussian anomalies used in the spike-tests of the P-wave model are larger than those of S-wavespeeds and density. Is it a plotting issue (or illusion?) or any specific reason for choosing it to be like that? It would be more insightful to show the spike-test results at other depths as well.

Indeed, there seemed to be a plotting issue. The figure has been adapted, and the supplementary material now contains further slices at 50, 300, and 500 km depth.

The authors have performed ~95 iterations, plus five additional iterations for every parameter during the spike tests. It is a large number of iterations. How expensive is each iteration? Or what is the overall computational cost?

In Table 1, the column *nx\*ny\*nz\*nt* in is a measure of computational cost per simulation. As the computational cost of waveform simulations scales to the fourth power of the maximum frequency, the bulk of the computational cost is in the final iterations; the first 50 iterations therefore represent less than ¼ of the total computational cost. Giving an absolute number of wall-clock time or GPU time is less informative, because it is very much dependent on the machine the simulations are run on. The same holds for e.g. (estimates of) energy consumption. The cluster that was used in our case was upgraded halfway through the work, which means that 'before' and 'after' values would not be comparable.

---

## Author Comment (AC2) · 21 Feb 2020

**Response to Reviewer 2**

Comments by Reviewer 2 in purple, answers in black.

(1) The authors claim that they also include radial anisotropy in their inversion, however, I cannot find any results for radial anisotropy in this manuscript. I strongly recommend the authors to present these solutions in order to make the statement consistent.

Information on the different Vsh and Vsv models is already presented in Figure 8 in the paper.

(2) It seems to me that the ETH group has published a number of papers on the same region (Europe) using the same approach (FWI). Why publishing another one or performing another inversion for the same target? What is the long term goal of this group for imaging the European upper mantle using FWI? Just including more data and publishing another paper once a while is not a good scientific exploration as far as I consider. For instance, I can list some papers which are similar to this manuscript. Can the authors provide some statements about significant improvements of this work in comparison to their previous work? Such as (1) Fichtner and Villasenor, 2015, EPSL; (2) Rickers et al, 2013, EPSL; (3) Fichtner et al, 2013, EPSL.

First, we note that science generally has two components: the development of new methods, and their application to new data. Both modes of operation are legitimate and needed to ensure progress, and one is not *per se* better than the other.

Second, we would like to point out that none of the publications listed above covers the region studied here. Fichtner & Villasenor (EPSL, 2015) considered the Western Mediterranean, with a focus more than 2000 km away from the current one. Rickers at al. (EPSL, 2013) studied the North Atlantic, with a focus on Iceland and Jan Mayen, more than 3000 km to the north of this study. Finally, Fichtner et al. (EPSL, 2013) focus on the Anatolian micro-plate and the North Anatolian Fault zone, to the east of the current study region. The work of Fichtner et al. (GJI, 2013), not listed above, covers all of Europe and Western Asia. Though this study includes the Eastern Mediterranean, it uses only around half of the amount of data for a model volume that is more than 100 times larger. Thus, Fichtner et al. (GJI, 2013) operated at a completely different scale in all respects.

This said, it is clear that the present study is entirely independent from the ones we did before, and it focuses on a region that had not been studied with the help of full-waveform inversion. Given the geologic relevance of the Eastern Mediterranean, we believe that our work is therefore more than justified.

(3) In fact, From Figures 7 and 14, I am surprised to see that predicted seismograms from their starting model (Fichtner et al, 2013) have such large mismatches in comparison to observed waveforms, even for this long-period frequency band. If their previous model is already good enough, we should not see such significant differences.

As explained in the response to the previous question, Fichtner et al. (GJI 2013) used a significantly smaller amount of data for a model volume that was orders of magnitude larger. Furthermore, the shortest period in Fichtner et al. (GJI 2013) was around 50 s, i.e., around twice as large as the shortest period used here. In summary, it is not at all surprising that the Eastern

Mediterranean part of the 2013 model does not explain the shorter-period data at much smaller scale considered here. This high misfit immediately serves as another motivation to carry out this work – clearly the existing models are insufficient.

(4) The authors just present their own tomographic model. In fact, the Mediterranean region is well studied by using either travel time tomography or even full waveform inversion by other groups. The authors ignored the same endeavor towards FWI imaging for the European crust and upper mantle by other groups. I wonder how their current model compares with other previous tomographic or FWI models? As far as I can tell from their Figure 12, the resolution for subducting slabs underneath the Mediterranean is quite low in comparison to previous travel time tomography (Wortel and Spakman, 2000, Piromallo and Morelli, 2003) or FWI models (Zhu et al, a series of studies). I strongly suggest the authors to make some comparisons with these previous studies, in order to convince the readers about the advantages or disadvantages of their current inversion.

We have added additional figures to the Supplementary Material, where we compare our final model to the Utrecht model UU-P07 (Amaru, 2007; this model is the successor of the model used in Wortel & Spakman, 2000). There is a clear correspondence between the models, which is encouraging given the fact that we compare a travel-time P-velocity model and a waveform S-velocity model. As is visible from these figures, the models have similar levels of detail as our final model – something similar holds for the models of Piromallo & Morelli (2003) and Zhu et al (2015). It may have appeared that there is more detail in the models mentioned by the reviewer, because those are usually viewed on a larger scale – in particular UU-P07, which is a global model. The advantage of zooming in, as we do, is that there is a lot more information, and as a result, e.g. the amplitudes of the anomalies are stronger.

(5) The authors claim that their model has good resolution down to the mantle transition zone. However, in their horizontal cross sections (Figure 10), I only see results down to 300 km. I suggest they to put more slices, say from 400 to 700 km.

We do not claim they have good resolution down to the mantle transition zone – only in some parts is the model updated down to the mantle transition zone. This is already summarised in Figure 8 of the paper, and we have now added further depth slices through both the final model and the resolution tests in the Supplementary Material.

(6) For vertical cross sections (Figures 12 to 13). I don't see any reasons to put slices with absolute velocity here. Since they will be dominated by 1D structures, in particular for the mantle transition zone, which involves velocity jumps across 410 and 660. I suggest only show relative velocity perturbations in their right panel.

One of the largest added benefits of using waveform tomography is the fact that we get access to absolute velocities. Any thermodynamical interpretation relies heavily on absolute velocity – see for example Deschamps & Trampert, 2003; Cobden et al, 2008, 2009, 2018. For this reason we prefer to show both types of cross-section.

(7) In fact, for the Hellenic subduction (Figure 13), numerous previous studies have demonstrated that the Hellenic slab can penetrate the 660 km discontinuity down to the lower mantle (from either travel time tomography or FWI), however, the authors' result seems different from previous conclusions. Do we need to re-evaluate this consensus, or the authors' model does

not have enough resolution to resolve it?

We agree with the reviewer that the initial formulation here was unfortunate; by no means did we intend to imply that the Hellenic slab must terminate near the MTZ! We have rephrased this part of the text to clarify this.

(8) For waveform comparisons, such as Figures 7 and 14, the authors only show comparisons between observed and predicted seismograms for a couple of station-source pairs. However, as pointed out by the authors, there are numerous seismograms. I am not surprised to see some improvements for some selected pairs of earthquakes and stations. How about other pairs? Since there are many stations in the authors' dataset, why not plot all three components (vertical, radial and transverse) for most data and predictions, just like common-shot gathers as did by exploration seismologists. I think that will be a better way to convince the readers that their model can really explain data, instead of just intensionally or randomly selecting some paris with good match. In addition, I suggest the authors to change color for predicted seismograms from the starting model, such as blue or other colors different from the current waveforms.

We have experimented with this. In the end we find that a comparison of single traces give the most insightful results, as is commonly done in waveform tomography. A record section, for our setup, results in the interference of too many different phases that are a result of significant 3-D structure. A record section may be more suitable in a situation where all receivers are located roughly in a line, as is the case for e.g. Beller et al (GJI, 2017). As a compromise we have included figure 6 which shows a very detailed misfit development.

(9) Figure 3, why changing iteration numbers for different frequency bands? In order to demonstrate that their model can fit data for different frequency ranges, I suggest just compare the starting one with their current model (100th iteration).

Figure 3 is meant as a demonstration of the multiscale method, and shows that the chosen points to include higher frequency content are suitable. This is why we prefer to show the iterations as given in the figure.

(10) Figure 4, it seems to me that the color of panel d is highly saturated, then they can see a strong banana kernel, how about using some reasonable colors. In addition, do they need to add additional weighting coefficients in order to use both body and surface waves for this inversion?

The colour scale in panel d is intentionally made the same as the colour scale in panels e and i. This oversaturation is the only way to make the kernels for all windows visible. This is intended as an illustration of the power of the windowing strategy: because of the significant phase shift visible in the body waves, the gradient in vp is much stronger for this window than for the surface wave window. This information would be lost almost entirely if a single window were taken for both arrivals. No additional weighting is included – this happens automatically through windowing.

(11) Figure 11, I suggest to plot iso-surface for relative velocity perturbation instead of absolute values, which is a typical way to examine subducting slabs.

With waveform tomography, we have a method that comes with the luxury of giving us absolute velocities. We are taking advantage of that because it directly feeds into thermochemical interpretation – this is the reason that we show these. Moreover, compared to the anomalies, the

background variations are relatively minimal, so showing relative instead of absolute velocities will not make a very large difference.

(12) In section 6.2, in order to evaluate their model, the authors chose additional 6 earthquakes as an independent dataset. As far as I can tell, this is a very small dataset to convince the readers that their model can explain waveforms that are not used in the inversion. I strongly suggest them to increase the size of this independent dataset.

We used events that cover the study area well. We would understand the argument if the events would cluster, but they don't. Doing more would just inflate the manuscript and result in unnecessary additional computational cost.

Some minor modifications: [...]

We have implemented the suggestions given by the reviewer.

---

## Editor Comment (EC1) · Caroline Beghein (Editor) · 25 Feb 2020

Dear authors,

After reading your response to reviewer 2's comments, I would like to echo their first comment. I understand the Vsv and Vsh models are shown in Figure 8. However, the anisotropy itself is not discussed in the paper, but I believe a (short) discussion of how it could be interpreted and its resolution would be valuable since it is a product of the inversion.

Best regards,

[Figure]

Caroline Beghein

---

## Author Comment (AC3) · 6 Mar 2020

In the manuscript we have now included a further discussion of anisotropy. We were initially hesitant to provide a further discussion of anisotropy, because the sensitivities to the different parameters vary substantially, and it is fundamentally impossible to separate intrinsic and extrinsic anisotropy. We have now adapted Figure 8 and included a new figure with anisotropy depth slices. We have also included some discussion in the text including the above caveats, which hope this provides a good balance.

[Figure]

[Figure]

**Fig. 1.** Updated Figure 8, including a panel with depth averaged anisotropy

[Figure]

**Fig. 2.** New anisotropy figure (new Fig. 11).